# Investigating Knee Joint Proprioception and Its Impact on Limits of Stability Using Dynamic Posturography in Individuals with Bilateral Knee Osteoarthritis—A Cross-Sectional Study of Comparisons and Correlations

**DOI:** 10.3390/jcm12082764

**Published:** 2023-04-07

**Authors:** Abdullah Raizah, Ravi Shankar Reddy, Mastour Saeed Alshahrani, Jaya Shanker Tedla, Snehil Dixit, Kumar Gular, Ajay Prashad Gautam, Irshad Ahmad, Praveen Kumar Kandakurti

**Affiliations:** 1Department of Orthopaedics, College of Medicine, King Khalid University, Abha 61421, Saudi Arabia; 2Department of Medical Rehabilitation Sciences, College of Applied Medical Sciences, King Khalid University, Abha 61421, Saudi Arabia; 3College of Health Sciences, Gulf Medical University, Ajman 4184, United Arab Emirates

**Keywords:** knee osteoarthritis, limits of stability, joint position sense, posturography, kinesthesis

## Abstract

(1) Background: Proprioception and limits of stability can significantly impact static and dynamic balance. Knee proprioception and limits of stability may be impaired in individuals with knee osteoarthritis (KOA). Impaired knee proprioception may impact the limits of stability, and understanding the associations between these factors is important for formulating treatment strategies in this population. The objectives of this study are to (a) compare the knee joint position error (JPE) and limits of stability between KOA and asymptomatic individuals and (b) assess the correlation between knee JPE and the limits of stability in KOA individuals. (2) Methods: This cross-sectional study included 50 individuals diagnosed with bilateral KOA and 50 asymptomatic individuals. Knee JPE was measured using a dual digital inclinometer at 25° and 45° of knee flexion (in the dominant and nondominant legs). The limits of stability variables, including reaction time (s), maximum excursion (%), and direction control (%), were evaluated using computerized dynamic posturography. (3) Results: The magnitude of the mean knee JPE is significantly larger in KOA individuals (*p* < 0.001) compared to asymptomatic individuals assessed at 25° and 45° of knee flexion in both the dominant and nondominant legs. The limits of stability test showed that KOA group individuals had a longer reaction time (1.64 ± 0.30 s) and reduced maximum excursion (4.37 ± 0.45) and direction control (78.42 ± 5.47) percentages compared to the asymptomatic group (reaction time = 0.89 ± 0.29, maximum excursion = 5.25 ± 1.34, direction control = 87.50 ± 4.49). Knee JPE showed moderate to strong correlations with the reaction time (r = 0.60 to 0.68, *p* < 0.001), maximum excursion (r = −0.28 to −0.38, *p* < 0.001) and direction control (r = −0.59 to −0.65, *p* < 0.001) parameters in the limits of stability test. (4) Conclusions: Knee proprioception and limits of stability are impaired in KOA individuals compared to asymptomatic individuals, and knee JPE showed significant relationships with the limits of stability variables. These factors and correlations may be considered when evaluating and developing treatment strategies for KOA patients.

## 1. Introduction

Knee osteoarthritis (KOA) is a degenerative joint disease commonly seen in the general population aged over 45 years [1]. It typically occurs as a result of subchondral erosion along with cartilage loss or thinning [2]. Individuals with KOA experience significant disability and impairment, leading to decreased quality of life [3]. Muscle weakness, ligament laxity, meniscal injury, and neuropathic pain are documented in KOA individuals, leading to increased pain with swelling, stiffness, and decreased range of motion [4]. Elderly KOA individuals experience diminished physical functioning, which hinders their social participation and increases their frequency of falls [4].

The proprioceptive system plays a crucial role in maintaining bodily balance, working closely with the visual and the vestibular systems [5]. Impaired proprioception can significantly affect somatosensory and motor control around the joint [6,7]. Knee joint proprioception integrates sensory input from a range of afferent receptors, which includes motion sense and joint sense [8]. Different authors have demonstrated increased knee joint positioning errors (JPE) in KOA individuals compared to age-matched asymptomatic individuals [9,10,11,12]. Different factors, such as degeneration, decreased muscle strength, and endurance, impair mechanoreceptor function, and thereby affect afferent proprioceptive input to the higher centers, resulting in altered motor output and proprioceptive function [13,14].

The limits of stability may be the most utilized procedure for assessing balance in a dynamic situation [15]. They quantify an individual’s ability to move their center of gravity to the limits of their stability without losing balance [16]. This protocol gives information on voluntary motor control, which also aids in the screening of fall risk in the elderly [16]. The normal sway angles in the anteroposterior (A-P) and mediolateral (M-L) directions are around 12.5 degrees and 16 degrees, respectively [16,17]. This region of stable oscillation is commonly known as the “Cone of Stability” [18]. The bounds of this stability cone vary continuously based on the activity being performed [18]. The limits of stability help in assessing balance in a dynamic situation by tracking the instantaneous change in COM velocity and position [19]. The limits of stability assess postural instability and identify persons at increased risk of falling [19]. Individuals with lower limits of stability have a greater risk of falling while shifting their body weight forward, backward, or from side to side, and are thus more susceptible to injuries [20]. The limits of stability are compromised in individuals with KOA, and this can lead to an increased risk of falling. Park et al. [21] examined the limits of stability between symptomatic KOA and healthy controls in a case–control study, and discovered that the limits of stability were impaired in symptomatic KOA patients. Park et al. [21]’s study had a sample size of only 14 diagnosed KOA individuals, and the authors recommended reproducing the study with a larger sample to see if similar results were observed.

The limits of stability are important in KOA individuals to maintain a dynamic balance [22]. It is postulated that impaired joint position sense can reduce postural control, limit stability, and cause individuals to fall [23]. As knee proprioception, postural control, and limits of stability are significant factors that can significantly cause impairment, disability, and falls in individuals with KOA [21,24], it is essential to determine the magnitude of knee JPE and impairment to the limits of stability in this population. Limited research exists that aims to determine whether a correlation exists between knee proprioceptive dysfunction and the limits of stability in people with KOA, or how strong that correlation might be. When evaluating patients with KOA and devising treatment regimens for these individuals, an understanding of the relationship between the two factors may be useful for clinicians or rehabilitation therapists. The objectives of this study are to (1) compare knee JPE and the limits of stability between KOA individuals and asymptomatic individuals and (2) assess the relationship between knee JPE and the limits of stability in KOA individuals. We hypothesize that knee proprioception and the limits of stability will be significantly impaired in KOA individuals compared to asymptomatic individuals, and that knee JPE may have a significant positive association with the limits of stability.

## 2. Materials and Methods

### 2.1. Design, Participants, Setting, and Ethics Statement

This comparative cross-sectional study included 50 participants diagnosed with KOA (mean age = 67.10 ± 4.36 years) and 50 asymptomatic participants (mean age = 66.50 ± 3.63). The study was carried out in the department of physiotherapy of the advanced research laboratory, Al-Farah, King Khalid University, Saudi Arabia. The study protocol was approved by the Institutional Ethical Committee, KKU (REC#12-118-2022). The symptomatic KOA individuals included participants who: (1) were over 45 years old, (2) had Kellgren–Lawrence (KL) grades between 2 and 4 on their knee radiographs [25], and (3) were able to stand independently. The subjects were excluded if they: (1) had surgery on their lower extremities, (2) had any injuries to their lower extremities, (3) had any neurological problems, (4) had a rheumatic disease, (5) had a history of cardiorespiratory instability, or (6) were not able to understand and follow the commands of the examiner. In this study, symptomatic KOA was defined as the presence of morning stiffness and knee pain with a visual analog score of >3. Asymptomatic adults were recruited for the study by distributing pamphlets and delivering lectures on the university campus and in nearby communities. Participants met the inclusion criteria if they were healthy, aged between 40 and 80, and able to comprehend and follow the examiner’s instructions. The study adhered to the ethical standards of the Declaration of Helsinki guidelines. Before the initiation of this investigation, all participants gave their consent.

### 2.2. Assessments and Outcomes

The demographic and functional assessments were performed by an expert physical therapist with 5 years of post-PhD experience in musculoskeletal physiotherapy.

#### 2.2.1. Knee Pain Severity

The current level of knee pain intensity was assessed using the VAS [26], which is a 100 mm horizontal line that has a beginning point indicating “no pain” and an endpoint indicating the “worst imaginable pain”. Each individual marked their current level of knee pain on the scale, and the examiner estimated the severity. The VAS is a reliable and valid tool to measure pain mechanics in KOA [26,27].

#### 2.2.2. Knee Society Score (KSS)

KOA individuals’ knee pain, function, and mobility were assessed using the knee society score (KSS) [28]. The scores ranged between 0 and 100, with a higher score indicating a higher level of function [28]. The KSS has two aspects, including a clinical score (KSS-C) and a functional score (KSS-F) [29]. The KSS is reliable, consistent, feasible, and has a discriminating ability in KOA individuals [30].

#### 2.2.3. Knee Joint Position Sense

The knee joint position sense was evaluated using a dual pro digital inclinometer (Dualer IQ, J-tech Medical, Midvale, UT, USA). A single investigator performed all the knee joint position sense assessments in both the dominant and nondominant extremities. All the evaluations were performed in a well-ventilated and quiet environment. All the subjects were asked to close their eyes throughout the joint position sense testing phase to eliminate visual input. The study adopted the active target reposition method to estimate the knee joint position sense [31]. The examiner defined the target positions as 25° and 40° of the knee flexion [32]. We choose these angles as these ranges represent the proprioceptive afferent input during the normal walking pattern as a measure of functional measurement [33]. To begin the knee joint position sense testing, the subject sat comfortably on a couch with their hip and knee flexed at 90°. One part of the dual inclinometer (the secondary inclinometer) was placed on the lateral aspect on the lower third of the femur along the joint line, and the second part (the primary inclinometer) was placed on the lateral aspect on the upper one-third of the fibula along the joint line, and secured using Velcro (Figure 1).

The examiner guided the participant’s leg from the starting position (90° of knee flexion) to the target position of 25° or 40° of knee flexion by extending the knee and placing it for a period of five seconds and asking them to remember this target position. Next, the examiner returned the participant’s leg to the starting position. Following this, the participant actively extended the knee to reposition to the target position as accurately as possible. The participant indicated by saying “YES” when they thought they had reached the target, and the reposition accuracy was measured as the joint position error (JPE) in degrees, as displayed on the digital inclinometer screen. Each of these procedures was repeated thrice, and the difference between the actual target position angle and the angle reproduced as sensed by the participant was recorded as the absolute error. We considered the joint JPE as the arithmetic mean of the absolute error. JPE = (trial one + trial 2 + trial 3/3).

#### 2.2.4. Limits of Stability

The limits of stability were assessed using computerized dynamic posturography (Iso-free, Techno body, Bergamo, Italy). The limits of stability comprise the area over which a subject can move safely without changing their base of support [21]. They consist of a circular platform and the following primary elements: a stabilometric posture platform, a touch screen, a 3D camera, and specific software [34]. The stabilometric force platform analyses the center of pressure when the individual is in the standing posture, sensing the pressure from the platform [34].

All the limits of stability assessments were performed in a calm and well-ventilated environment. Each individual was asked to stand on the stabilometric force platform with both feet together in a standardized manner (Figure 2).

The subject was asked to look at the screen in front of them and follow the targets provided on the screen by the posturography device intended to assess the limits of stability. Only once did the target randomly appear in all eight directions, as shown on the screen in a blink. Without moving their feet, the individuals were told to shift their center of mass toward the objective (Figure 3).

The instrument recorded and provided a score for the amount of sway needed to reach the target from the center along the exact shortest vertical or horizontal path. One hundred was the maximum score that could be obtained in a direction. A lower score indicated greater sway. The limits of stability parameters included were reaction time, maximum excursion, and directional control [35].

#### 2.2.5. Reaction Time

The reaction time was calculated as the ability of the individual reacting to reach the target position by initiating voluntary movement in response to the stimulus provided on the computer screen [35]. The beginning of a subject’s voluntary shifting was defined as the instant in which the normalized center of pressure reached an amplitude that was higher than the peak amplitude of the normalized center of pressure recorded, over the course of a control period that lasted for two seconds, before the response signal [35].

#### 2.2.6. Maximum Excursion

To determine the respondents’ maximal excursion, they were asked to lean as far as they could toward one of the eight randomly assigned spatial target places while maintaining 100% of their stability [35]. The maximum distance covered by the normalized center of pressure was measured by the posturography device [35].

#### 2.2.7. Directional Control

To establish directional control, the amount of movement of normalized COP in the target direction (in the direction that leads toward the target) was contrasted with the amount of movement in the off-target direction [35]. Its value was calculated as the difference between the normalized COP’s on-target movement and off-target movement, represented as a percentage of the overall on-target movement, as shown in the formula: [(amount of on-target movement—the amount of off-target movement)/(amount of on-target movement)] × 100 [35]. A straight path resulted in no off-target movement and a score of 100% for directional control. The computerized dynamic posturography equipment’s algorithm was used to determine the direction of control.

### 2.3. Statistical Analysis

Shapiro–Wilk tests were used to assess the normality of the study variables, and the study variables followed a normal distribution. Differences in knee JPE and limits of stability characteristics between the KOA and asymptomatic groups were analyzed using ANOVA (One-Factor ANOVA). Correlations between the Knee JPE and limits of stability were determined using Pearson’s correlation coefficient (r). The correlations were estimated to be low (r = 0.20 to 0.30), moderate (r = 0.31 to 0.69), or high (r = 0.70 to 1) [36]. Statistical analyses were performed using the Statistical Package for the Social Sciences (SPSS.22, IBM SPSS Statistics for Windows, Version 22.0 Armonk, NY, USA: IBM Corp.). A two-tailed *p* value < 0.05 was regarded as statistically significant.

## 3. Results

This cross-sectional study included 50 participants with KOA (mean age: 67.10 ± 4.36 years) and 50 asymptomatic participants (mean age: 66.50 ± 3.63 years). There were no differences in age, gender, height, weight, or body mass index between the groups (Table 1).

The magnitude of the mean JPE in the KOA group is larger (dominant: 25° of flexion: F = 53.50, *p* < 0.001; dominant: 45° of flexion: F = 49.00, *p* < 0.001; nondominant: 25° of flexion: F = 77.47, *p* < 0.001; nondominant: 45° of flexion: F = 90.66, *p* < 0.001) compared to the asymptomatic group. The mean knee JPE in the KOA group ranged from 5.40° (dominant: 40 of flexion) to 5.85° (nondominant: 40 of flexion) and from 3.44° (nondominant: 40 of flexion) to 3.68° (nondominant: 25 of flexion) in the asymptomatic group. The limits of stability were compromised in the KOA group compared to the asymptomatic group, as summarized in Table 2 and Figure 4.

The limits of stability tests showed that KOA group individuals had a longer reaction time (1.64 ± 0.30 s) and reduced maximum excursion (4.37 ± 0.45) and direction control (78.42 ± 5.47) percentages compared to the asymptomatic group (reaction time = 0.89 ± 0.29, maximum excursion = 5.25 ± 1.34, direction control = 87.50 ± 4.49).

Knee JPE showed significant correlations with the limits of stability variables (Table 3 and Figure 5, Figure 6 and Figure 7).

Knee proprioception showed statistically significant moderate to strong correlations with the reaction time (r = 0.60 to 0.68, *p* < 0.001), maximum excursion (r = −0.28 to −0.38, *p* < 0.001), and direction control (r = −0.59 to −0.65, *p* < 0.001) parameters of the limits of stability test.

## 4. Discussion

We conducted this comparative cross-sectional study to compare knee JPE and limits of stability variables between KOA and asymptomatic individuals. Additionally, we assessed the relationship between knee JPE and limits of stability variables in KOA individuals. The results of this study showed that knee proprioception and the limits of stability are impaired in KOA individuals compared to asymptomatic individuals, and those in individuals with knee JPE showed a significant moderate to a strong relationship with the limits of stability variables.

This study’s findings, which show substantial proprioceptive impairment in terms of increased JPE in the KOA group, are in accordance with findings from previously published papers that involve subjects with other types of knee condition [37,38]. Degenerative changes or atrophy that weaken muscles can damage joint structures, affect mechanoreceptors and proprioceptive afferent input to the higher centers, and compromise knee proprioception, which are a few potential causes of decreased proprioception in KOA patients [10,39,40]. Proprioception and the intricate afferent performance of the proprioceptive network system are affected by pain and edema in KOA patients [10,40]. Alshahrani et al. [10] showed that the magnitude of knee JPE was larger in individuals with KOA compared to asymptomatic groups in sitting positions (20° of flexion: d = 1.73, *p* < 0.001; 40° of flexion: d = 1.72, *p* < 0.001; 60° of flexion: d = 1.83, *p* < 0.001) and standing positions (20° of flexion: d = 1.89, *p* < 0.001). The mean knee JPE in our study ranged from 5.4° to 5.8°, which is the average error shown for KOA in different studies [10,11,40,41,42].

Regarding the limits of stability, they quantify movement qualities associated with participants’ ability to voluntarily alter their spatial position and their capacity to maintain stability in that posture [35]. Several studies have elucidated that postural instability and balance decreases in KOA individuals [11,43,44]. The current study’s findings on the limits of stability in double stance support earlier findings that patients with knee osteoarthritis perform worse than healthy individuals in trials with both open and closed eyes when posturography is used to determine postural stability [45]. The limits of stability may be impacted by the interplay between muscle changes and proprioceptive inaccuracy in KOA individuals. Muscle changes include having less muscle mass, incomplete muscle activation, lower muscle spindle sensitivity, and fewer sensory units (i.e., fewer mechanoreceptors) [38,46,47]. Regarding the muscles around the knee, changes in the cross-sectional area and fatty degeneration of the knee muscles due to arthritis have been reported [48]. Additionally, decreased knee proprioceptive capacity will have an effect on the limits of stability in KOA individuals [13], since reduced proprioceptive afferent input to higher centers will modify the peripheral motor output, resulting in impaired somatomotor control [49]. Sanchez-Ramirez et al. [50] showed that postural control is altered and proprioceptive accuracy is impaired in 284 KOA individuals. According to Ye, Jiajia et al. [41], the KOA group had greater anterior-to-posterior and medial-to-lateral sway velocities affecting postural control. 

This study showed a significant correlation between knee JPE and the limits of stability variables. Larger proprioceptive errors are correlated with increased reaction time, slower maximum excursion, and direction control parameters. Lord and colleagues [51] showed a significant correlation between sway velocities and knee active repositioning test results (r = 0.43, *p* < 0.005). Additionally, Hurley et al. [52] showed a higher correlation (r = 0.53, *p* < 0.001) between increased knee JPE and lower excursion values. These results and those of the current investigation show that knee joint proprioception has a more significant role in the regulation of dynamic standing balance than static standing balance, which is necessary for various functional activities involving weight shifting. Knee JPE and reaction time had a positive association with each other (r = 0.60 to 0.68, *p* < 0.001), which suggests that KOA participants responded to a visual stimulus more slowly as knee proprioception was significantly impaired. Lord et al. [53] conducted a study on older participants to step on one of four targets as rapidly as they could when it was illuminated to test the individuals’ reaction times. Further analysis of our findings revealed a negative relationship between knee JPE and the maximum excursion (−0.28 to −0.38, *p* < 0.001) and directional control (−0.59 to −0.65, *p* < 0.001) parameters. In other words, participants with better knee joint position sensing acuity were able to increase their stability thresholds. Prior research demonstrated that the limits of stability decreased with the degeneration process, and were a key predictor of multiple falls [21,54]. When compared to participants who had never fallen before, they discovered that fallers had significantly longer reaction times and lower maximum excursion and direction control parameters. The current study generates a hypothesis that states that enhancing knee proprioception may enhance reaction time and minimize the occurrence of falls among the elderly; this must be investigated in future studies.

Clinical practice must be changed to include ongoing evaluations of limits of stability and proprioception in patients with knee OA, especially in those patients who choose conservative treatment over surgical intervention, in light of the impairments in postural stability and proprioception found in this study. These observations are further supported by the finding that proprioception and postural stability measurements have a connection to clinical and functional ratings. So, it appears that putting into practice specialized rehabilitation strategies is of utmost importance to enhance proprioception and postural stability [55] and evaluate their efficacy in improving clinical and functional scores. From a therapeutic standpoint, our findings imply that proprioceptive precision and the limits of stability variables of postural control are associated. Hence, rehabilitation with an emphasis on knee proprioception and neuromuscular factors may enhance postural control in KOA patients. To confirm this, additional intervention research is required. Adopting the Iso-free technology posturography system as an evaluation tool, a straightforward clinical test to evaluate postural control in patients with KOA, is a key strength of our study. Moreover, the inclusion of 50 KOA and 50 asymptomatic patients to assess knee JPE and the limits of stability to effectively evaluate our hypothesis is a further advantage of this study over the vast majority of previous investigations.

The Iso-free (techno-body) proprioceptive assessment system, a cutting-edge tool for measuring proprioception and postural stability, was used in the current investigation. It was proven to offer trustworthy metrics, as previously described in the literature [34]. The Iso-free technology system, in particular, enables estimation of the limits of stability component of postural stability when standing. The occurrence of high stability index values during the trial with open eyes are thought to be a sign of a more refined control, because they are an indication of efficient proprioceptive reflexes that may quickly stabilize the patient before the vestibular responses can be triggered [34]. Additionally, this is the most effective way of assessing knee proprioception utilizing an objective instrumental measure (a dual digital inclinometer) and the target reposition sense method when the eyes are closed. Compared to robust isokinetic or 3D computer fast-track analysis systems, digital inclinometers are more user-friendly, can be handled by a single rater, are reliable, and can be moved to the testing field [31,56,57,58].

### Limitations

We postulated that impaired postural control might result in activity restrictions, and that it may be influenced by muscular weakness, proprioceptive inaccuracy, and knee instability. However, the current study’s cross-sectional design can only demonstrate the existence of relationships, not their underlying causes. Our hypotheses must be supported by longitudinal investigations.

## 5. Conclusions

This study concluded that KOA individuals have impaired proprioception and limits of stability compared to asymptomatic individuals. Additionally, knee JPE is significantly associated with the limits of stability variables (reaction time, maximum excursion, and directional control). These factors and associations must be taken into account when evaluating and developing treatment strategies for KOA patients. Considering the relationship between proprioception, postural control, and frequency of falls, future research should investigate the effects of proprioceptive training programs on limits of stability variables and the frequency of falls in the elderly KOA population.

## Figures and Tables

**Figure 1 jcm-12-02764-f001:**
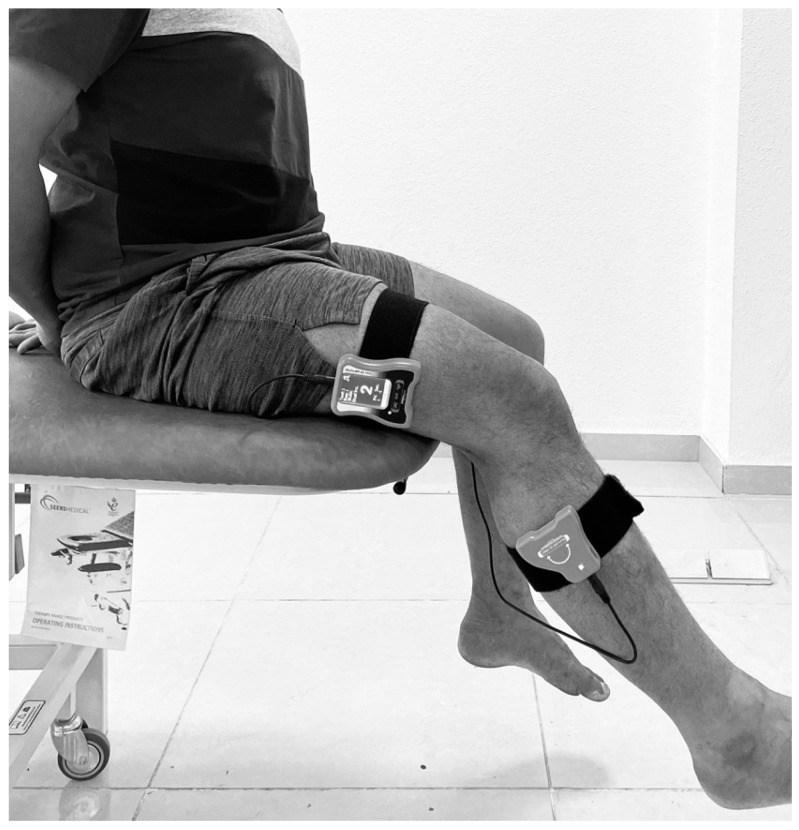
Knee joint position error assessment using a dual digital inclinometer.

**Figure 2 jcm-12-02764-f002:**
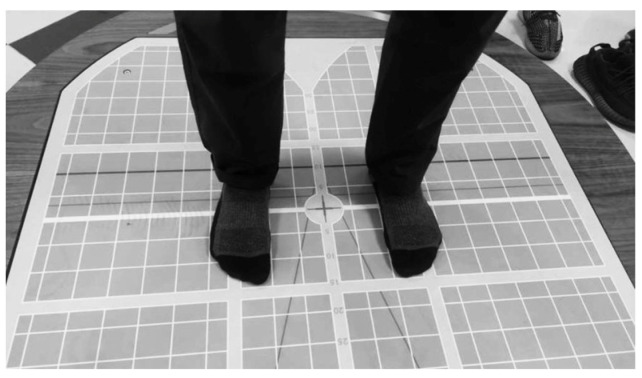
Individuals stood on a posturography force platform with both feet in the standard position.

**Figure 3 jcm-12-02764-f003:**
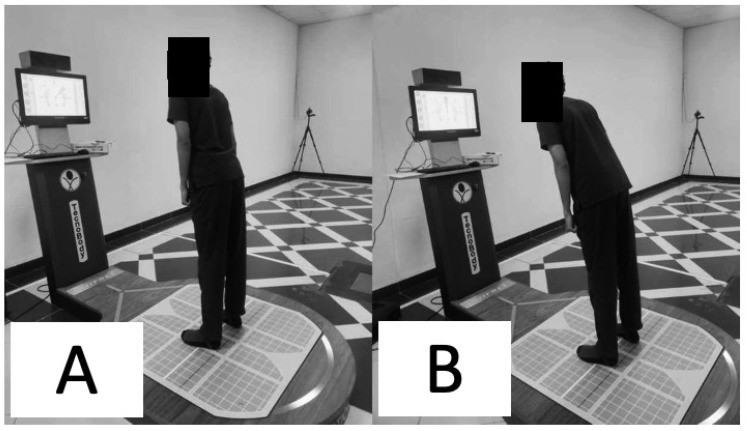
Limits of stability assessment (**A**) at the start and (**B**) when moving to a target.

**Figure 4 jcm-12-02764-f004:**
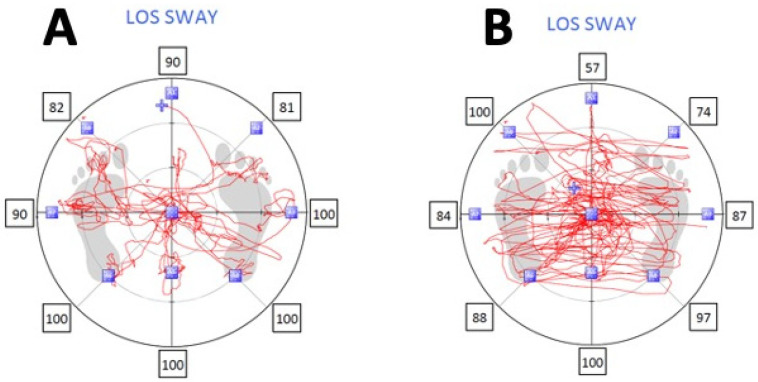
Limits of stability sway in (**A**) asymptomatic and (**B**) knee osteoarthritis groups.

**Figure 5 jcm-12-02764-f005:**
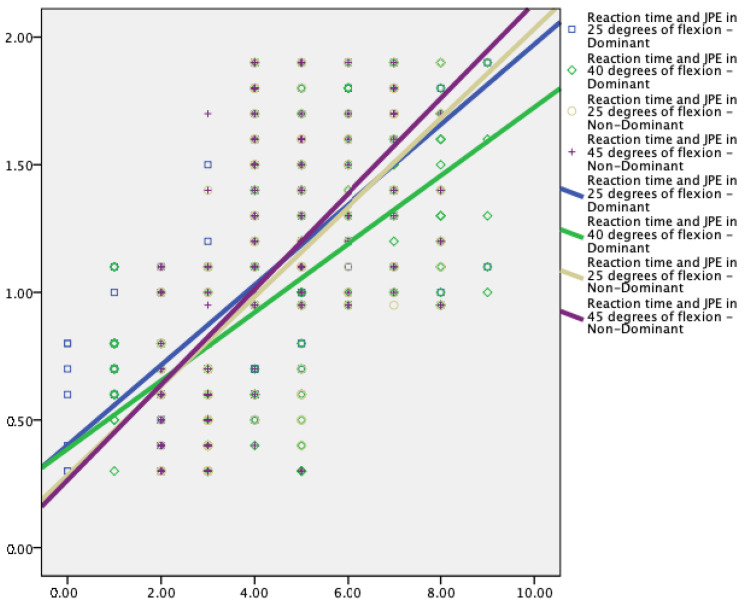
Relationship between Reaction time (s) and knee joint position error at 25° and 40° flexion in dominant and nondominant legs.

**Figure 6 jcm-12-02764-f006:**
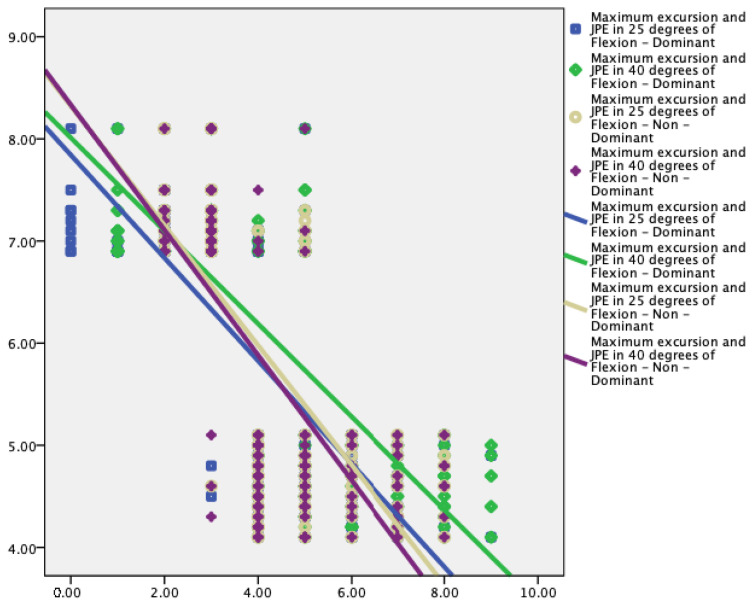
Relationship between maximum excursion (%) and knee joint position error at 25° and 40° flexion in dominant and nondominant legs.

**Figure 7 jcm-12-02764-f007:**
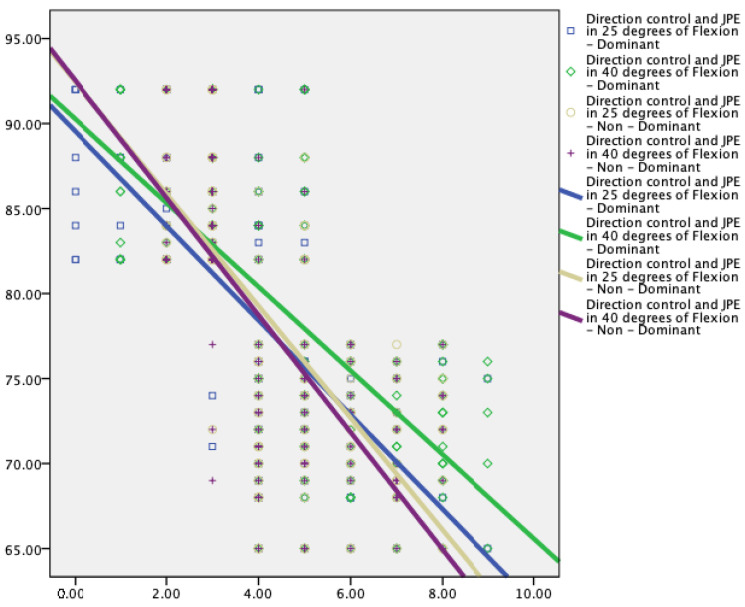
Relationship between direction control (%) and knee joint position error at 25° and 40° of flexion in dominant and nondominant legs.

**Table 1 jcm-12-02764-t001:** Physical and demographic characteristics of study participants.

Variables	Knee OA (*n* = 50)	Asymptomatic (*n* = 50)	*p*-Value
Age (years)	67.10 ± 4.36	66.50 ± 3.63	0.475
Gender (M:F)	28:22	32:18	0.143
Height (meters)	1.67 ± 0.09	1.73 ± 0.05	0.302
Weight (kg)	71.34 ± 5.96	70.58 ± 5.24	0.130
BMI (kg/m^2^)	24.40 ± 3.50	23.38 ± 2.14	0.441
Pain intensity (VAS score, 0–10)	5.99 ± 1.70	n/a	-
Knee Society knee score	48 ± 13	n/a	-

BMI = body mass index. The KOA group had significantly increased knee JPE compared to the asymptomatic group (Table 2).

**Table 2 jcm-12-02764-t002:** Comparison of knee joint position sense and limits of stability tests between knee OA and asymptomatic groups.

Variables	Asymptomatic (*n* = 50)	Knee OA (*n* = 50)	F	*p*-Value
Dominant—knee JPE at 25° of flexion (°)	3.56 ± 1.05	5.48 ± 1.53	53.50	<0.001
Dominant—knee JPE at 40° of flexion (°)	3.60 ± 0.86	5.40 ± 1.60	49.00	<0.001
Nondominant—knee JPE at 25° of flexion (°)	3.68 ± 0.79	5.83 ±1.53	77.47	<0.001
Nondominant—knee JPE at 40° of flexion (°)	3.44 ± 0.86	5.85 ± 1.57	90.66	<0.001
Reaction time (s)	0.89 ± 0.29	1.64 ±0.30	162.48	<0.001
Maximum excursion (%)	5.25 ± 1.34	4.37 ± 0.45	19.44	<0.001
Direction control (%)	87.50 ± 4.49	78.42 ± 5.47	82.36	<0.001

JPE = joint position error. p values are based on post hoc Bonferroni correction.

**Table 3 jcm-12-02764-t003:** Relationship between cervical joint position error and balance and limits of stability tests (*n* = 200).

Variables		Reaction Time (s)	Maximum Excursion (%)	Direction Control (%)
Dominant—Knee JPE at 25° of flexion (°)	r	0.68 **	−0.28 **	−0.60 **
Dominant—Knee JPE at 40° of flexion (°)	r	0.60 **	−0.30 **	−0.59 **
Nondominant—Knee JPE at 25° of flexion (°)	r	0.66 **	−0.34 **	−0.62 **
Non- Dominant—Knee JPE at 40° of flexion (°)	r	0.68 **	−0.38 **	−0.65 **

JPE = joint position error. ** = Correlation is significant at the 0.01 level (2-tailed).

## Data Availability

All data generated or analyzed during this study are included in Appendix A.

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
