# Peer review of "Investigating Knee Joint Proprioception and Its Impact on Limits of Stability Using Dynamic Posturography in Individuals with Bilateral Knee Osteoarthritis—A Cross-Sectional Study of Comparisons and Correlations"

_jcm, 2023, doi:10.3390/jcm12082764_

Round 1

Reviewer 1 Report

I thank the authors for having addressed this research topic. KOA still remains a topical subject among the knee osteoarticular pathologies. Indeed, this is a cross-sectional study on knee proprioception and its impact on postural stability in patients with bilateral KOA.

The study is well conducted and the article is methodologically well written.

Below are few comments :

Abstract: 

- Please remove the subtitles (Background, Materials and Methods..etc.). The abstract should be in one big compact paragraph (c.f. Articles in the web site of the journal).

Statistical Analysis

Im asking about the post hoc test after doing comparison with ANOVA? 

Could you mention the established ANOVA plan? How many factors?

Reviewer 2 Report

Raizah et al. compared joint proprioception and stability between KOA patients and asymptomatic controls. These findings are not surprising and consistent with previous studies. Here are my comments:

1.     The first sentence of the introduction sounds strange - ‘Knee OA is … due to degenerative joint disease? ‘ The citation for this sentence is inappropriate as well.

2.     How did the authors define symptomatic?

3.     Did the authors assess knee pain in both sides, considering the participants had bilateral knee OA?

4.     Why were individuals with CVD excluded?

5.     Sample size calculation seems redundant in this exploratory study. The reason for selecting an effect size of 0.83 is unclear.

6.     In Figure 4, does it show that the stability scores of KOA patients are greater/better than asymptomatic controls in multiple directions?  This contrasts to the results shown in Table 2, where reaction time, maximum excursion, and direction control are worse in KOA patients.

7.     There are too many spelling and grammar errors. Language must be improved.

Author Response

Please find the attached word document. 

Round 2

Reviewer 2 Report

1. Some researchers consider knee OA as a degenerative joint disease, but no one says that knee OA is DUE TO degenerative joint disease.

2. It seems to me that the authors did not clearly define symptomatic knee OA. The response and added contents only explain what symptomatic knee OA is but not how it was defined in this study. For example, severity and duration of symptom are unclear.

3. The sample size calculation is not necessary for this study. If the authors insist to do sample size calculation, something should be clarified. The mean scores used for calculating sample size seem arbitrary, and some data are missing (e.g. SD and an MCID for JPE scores), in the current version.

Author Response

Please find the attached word document.
